# Comparative Validation of the fBrake Method with the Conventional Brake Efficiency Test Under UNE 26110 Using Roller Brake Tester Data

**DOI:** 10.3390/s25144522

**Published:** 2025-07-21

**Authors:** Víctor Romero-Gómez, José Luis San Román

**Affiliations:** 1Universidad Carlos III de Madrid, Departamento de Ingeniería Mecánica, Calculation and Transports (MECATRAN), Avenida de la Universidad, 30 (Edificio Sabatini), 28911 Leganés, Spain; jlsanro@ing.uc3m.es; 2Instituto de Seguridad de los Vehículos Automóviles (ISVA-UC3M), Avenida de la Universidad, 30 (Edificio Sabatini), 28911 Leganés, Spain; 3Mechanical and Civil Engineering Department, Purdue University Northwest, 2200 169th St., Hammond, IN 46323, USA

**Keywords:** sensor calibration, braking systems, uncertainty analysis, vehicle safety, vehicle inspection, roller brake tester

## Abstract

**Highlights:**

**What are the main findings?**
The fBrake method reliably estimates the braking efficiency of laden light passenger vehicles using measurements from unladen conditions.Comparative testing shows high agreement between the fBrake method and the conventional brake efficiency test, in compliance with UNE 26110 standard criteria.

**What is the implication of the main finding?**
PTI stations can adopt the fBrake method as an alternative when direct laden testing is impractical, maintaining regulatory equivalence.The validated approach improves the accuracy and practicality of brake inspections, supporting safer vehicle operation and regulatory compliance.

**Abstract:**

In periodic technical inspections (PTIs), evaluating the braking efficiency of light passenger vehicles at their Maximum Authorized Mass (MAM) presents a practical challenge, as bringing laden vehicles to inspection is often unfeasible due to logistical and infrastructure limitations. The fBrake method is proposed to overcome this issue by estimating braking efficiency at MAM based on measurements taken from vehicles in more accessible loading conditions. In this study, the fBrake method is validated by demonstrating the equivalence of its efficiency estimates extrapolated from two distinct configurations: an unladen state near the curb weight and a partially laden condition closer to MAM. Following the UNE 26110 standard (Road vehicles. Criteria for the assessment of the equivalence of braking efficiency test methods in relation to the methods defined in ISO 21069), roller brake tester measurements were used to obtain force data under both conditions. The analysis showed that the extrapolated efficiencies agree within combined uncertainty limits, with normalized errors below 1 in all segments tested. Confidence intervals were reduced by up to 74% after electronics update. These results confirm the reliability of the fBrake method for M1 and N1 vehicles and support its adoption as an equivalent procedure in compliance with UNE 26110, particularly when fully laden testing is impractical.

## 1. Introduction

The accurate assessment of braking efficiency is essential for ensuring vehicle safety and maintaining regulatory compliance during periodic technical inspections (PTIs) across the European Union [1]. Among the most widely used inspection tools, roller brake testers (RBTs) offer a practical means to evaluate braking force by simulating road conditions under controlled circumstances. However, the accuracy and repeatability of these systems depend on the quality and calibration of the embedded force sensors [2], making it essential to understand and control the associated uncertainty in brake force estimation.

Recent studies have explored various methods to improve the reliability of brake testing under PTI conditions. Škreblin et al. [2] proposed a methodology for calculating RBT measurement uncertainty, emphasizing metrological traceability in accordance with ISO/IEC 17025 [3] standards. Surblys and Sokolovskij [4], and later Šarkan et al. [5], highlighted the effects of vehicle loading, tire conditions, and axle distribution on braking test variability. In addition, Pexa et al. [6] experimentally assessed the use of small-diameter rollers for evaluating rolling resistance and braking behavior. While these studies primarily focus on the mechanical and metrological aspects of RBT platforms, Nieto et al. [7] advanced the discussion by proposing the fBrake model in the context of Spanish PTIs, particularly for semi-trailers, to align with European regulatory frameworks such as UN/ECE Regulation No. 13 [8], 2010/48/UE [9], and now 2014/45/UE [10].

In a previous study published in Sensors [11], we introduced the fBrake method as an alternative procedure for estimating the braking efficiency of laden light-duty and passenger vehicles using RBT measurements obtained in unladen or partially laden states. This approach addressed a longstanding limitation in PTIs: vehicles are typically tested without additional load, which can result in underestimating actual braking performance. The initial results demonstrated that the fBrake method could effectively simulate laden braking behavior while operating within the capabilities of standard RBT equipment.

However, at the time of that publication, the method had not yet been validated against official regulatory standards. In Spain, the UNE 26110 standard [12] (Road vehicles. Criteria for the assessment of the equivalence of braking efficiency test methods in relation to the methods defined in ISO 21069 [13,14]) defines a process for determining whether an alternative method can be considered equivalent to the conventional test. This includes guidelines for estimating measurement uncertainty using techniques such as resampling (e.g., Bootstrap) and establishing compatibility between results via normalized error analysis. Demonstrating that the fBrake method yields statistically equivalent results is crucial for its regulatory adoption and practical deployment in inspection stations.

The present study addresses this need by conducting a comparative experimental validation of the fBrake method in accordance with UNE 26110. Using calibrated RBT data from light vehicles under routine PTI conditions, we compare braking force estimates and their associated uncertainties across different loading configurations. Special attention is given to the role of sensor technology, as the accuracy and stability of load cells and control electronics are fundamental to the integrity of the measurement process.

This follow-up investigation provides new empirical support for the fBrake method as a viable tool for improving brake efficiency assessments in PTIs. Its implementation may reduce the need for invasive ballasting procedures or subjective interpretations of compliance, thereby enhancing cost effectiveness and fairness in inspection protocols, especially for M1 and N1 category vehicles, where full loading is often impractical.

## 2. Materials and Methods

The main equipment used for the experimental phase of this study is the roller brake tester (RBT) located at Universidad Carlos III de Madrid (make: CAPATEST (Barcelona, Spain); model: RBT-2000/F). To provide an overview of the experimental design, data analysis, and validation steps, a general methodology roadmap is presented in Figure 1. This diagram summarizes the key stages of the study, including vehicle testing, application of the fBrake method, uncertainty analysis, and the final validation process following UNE 26110 guidelines.

### 2.1. The Roller Brake Tester (RBT)

Roller brake testers (RBTs) are widely used diagnostic devices designed to evaluate the brake efficiency of road vehicles. These systems simulate real-world rolling conditions by allowing the wheels of the vehicle to rotate over powered rollers while the operator applies the brakes. The resulting braking forces are measured and analyzed to assess compliance with safety standards. RBTs are routinely used in vehicle inspection centers, mechanical workshops, and research laboratories, including ISVA-LABITV at Universidad Carlos III de Madrid.

The RBT employed in this study is a Capatest RBT-2000/F model, which features powered rollers and advanced data acquisition systems. The principle behind its operation is based on the friction force between the brake pads and the brake disc (or drum) and, subsequently, between the tire and the roller surface. The braking force F_b,i_ for each wheel is calculated as:(1)$$F_{b,i}=\mu \cdot N_{i}$$
where μ is the coefficient of friction, and N_i_ is the normal force on the i-th wheel. The resistance to roller rotation under braking generates a torque:(2)$$\tau =F_{b,i}\cdot r\;$$
where r is the effective radius of the roller.

#### 2.1.1. Main Components

Drive Rollers: These steel-core rollers, coated with high-friction surfaces, rotate at controlled speeds to simulate wheel movement. Depending on the design, RBTs can use either unpowered rollers—where the vehicle’s own motion drives the rollers—or powered rollers with electric or hydraulic motors. The RBT used in this research features powered rollers, enabling precise simulation of driving conditions and the ability to evaluate high braking forces.

Force Sensors: These measure the braking force transmitted by each wheel and are integrated within the roller assembly. Two types are commonly used:

Piezoelectric force transducers leverage the piezoelectric effect of non-centrosymmetric crystal materials (e.g., PZT and BaTiO_3_) to produce a voltage when mechanically stressed. These sensors provide fast and precise readings within a defined calibration range but are sensitive to temperature [15].

Strain gauge-based load cells use foil-type strain gauges configured in a Wheatstone bridge. They require more frequent recalibration but offer robust performance for continuous operation under variable thermal conditions. This sensor type is used in the Capatest RBT-2000/F.

Control System: The control unit governs roller motion, data acquisition, and analysis. It comprises an electronic control board connected via Ethernet to a computer system; all housed in a control cabinet. Safety features such as emergency stop buttons are included. The system software processes sensor data and computes parameters like braking imbalance, effectiveness, and disc ovality.

Tachometer Roller (spring-loaded sensor roller): An auxiliary free-spinning roller in contact with the test wheel measures its tangential velocity. Mounted on a single-degree-of-freedom support, it adapts to the wheel’s position and helps detect the onset of micro-slippage by comparing its speed to that of the drive rollers. The test ends when this slippage is detected.

Additional Components: These include the structural frame (embedded or elevated), load cell mounts, suspension test beds, cabling, chains, linkages, and other mechanical supports. The RBT at Universidad Carlos III de Madrid is installed on a raised platform integrated with weighing scales and suspension test equipment.

#### 2.1.2. Calibration

Once the main components of the brake tester are described, it is necessary to explain how to calibrate the equipment. This requires both mechanical calibration of the load cells and electronic calibration of the board that governs it, if it is being used for the first time. The arrangement of the rollers is as follows (Figure 2):

The red ellipse shows the location of the load cell measuring the left roller motor pitch, while the yellow ellipse shows the location of the load cell measuring the right roller motor pitch. By assembly scheme, since the movement of the rollers drives the vehicle downward in Figure 2, the left roller load cell measures the tension and the right roller load cell measures the compression.

Calibrating the left load cell (tensile) is simple to perform with any standard calibration fixture that has a base long enough to cover the distance between the platforms before and after the rollers, while the right cell is not possible without a specific fixture. Since it is necessary to compress, it is necessary to anchor the base of the fixture to the brake tester by means of screws in the holes marked in blue and to be able to compress by applying a torque to the right motor by anchoring it to the load cell (mounted on the fixture) by means of an adapter plate (Figure 3) placed in the position indicated by the white ellipse.

The tool modeled in SOLIDWORKS v.2023 to achieve the required calibration is shown on Figure 4.

#### 2.1.3. Repeatability Analysis

To determine whether the RBT, owned by the Department of Mechanical Engineering of UC3M, provided repeatable data, a preliminary test was conducted using a vehicle (make: Volkswagen; model: Scirocco MY2010). The braking force of the front axle was measured in a “driver-only” load configuration, as well in a partially loaded one.

Despite having calibrated the roller brake tester prior to the preliminary test, it was important to perform a repeatability analysis, as having an up-to-date calibration curve does not guarantee that the machine will produce reproducible measurements.

This test was performed using the original electronics of the roller brake tester (Make: CAPATEST (Barcelona, Spain), models: RBT 299 + PAD 224 connected by a flat ribbon AWM 2651 105C 300V VW-1 cable) linked via a USB type B to type A 2.0 cable (up to 480 Mbps over 5 m [16]) to a PC running Windows XP. Figure 5 shows the USB type B port highlighted in light blue.

As shown in Table 1, the roller brake tester exhibited significant repeatability issues, particularly in the loaded configuration. The standard deviation reached 120.8 daN on the left roller and 129.8 daN on the right, corresponding to coefficients of variation (CVs) of 39.17% and 40.67%, respectively. In contrast, under the unloaded configuration, the standard deviations dropped to 21.87 daN and 20.83 daN, with corresponding CVs of 5.49% and 4.99%. These normalized dispersion metrics highlight a much higher variability in the system response under higher braking forces, reinforcing the hypothesis that the system was electronically limited rather than mechanically unstable.

An analysis of potential causes ruled out a low slip threshold, which would have prematurely terminated all trials uniformly, and dismissed calibration or load cell faults, since the variability appeared in both channels simultaneously (see Figure 6). The results pointed to the outdated control electronics as the root cause, justifying the subsequent modernization phase aimed at improving signal stability and timing accuracy.

Despite the observed inconsistencies, a statistical study was conducted using the data from the five test repetitions. First, the raw data were analyzed separately for the left and right rollers; subsequently, Bootstrapping [17] was applied as a resampling method.

The parameters for validation (overall mean and the lower and upper bounds for a 95% confidence level) were calculated based on 10,000 Bootstrap samples that are represented in Figure 7. The upper and lower limits are determined as follows:(3)$$Upper\;Limit={Limit}_{r}\;$$(4)$$Lower\;Limit={Limit}_{\left(r+q\right)}$$
where(5)$$r=\frac{\left(n-q\right)}{2}\;$$(6)q=p·n(7)q=0.95

From the results obtained with the calibrated roller brake tester (Table 2), it was evident that although such testers are not inherently characterized by high repeatability, the data variability observed was excessive and strongly dependent on the timing of the test.

After investigating the factors contributing to this variability, it was found that the performance of the PC running the control software affected the repeatability of the measurements. Following consultation with the manufacturer, it was decided to modernize the entire electronic system before commencing the main series of tests for this research. The old dual-board system was replaced with a single, unified Make: CAPATEST (Barcelona, Spain), model: PAD-728 RTX board and a more modern PC featuring a fourth-generation Intel Core i7 processor and 8 GB of RAM, running Windows 10. Additionally, a dual RJ45 Ethernet port was installed to maintain an Internet connection for software updates while allowing a second Cat 6 Ethernet cable (10 Gbps, up to 55 m [18]) to link the new electronic board directly to the PC. The new system thus eliminates the USB connection in favor of Ethernet. Figure 8 highlights the new Ethernet port in blue.

Braking forces logged with the new electronics after the 5-test array, right before bootstrapping, are shown on Table 3 and Figure 9.

A markedly more stable sequence of data can be seen compared to the preliminary test. After the replacement of the legacy dual-board electronics with the modernized PAD-728 control unit, the system exhibited a marked improvement in measurement repeatability, particularly under the loaded test condition. In the post-upgrade tests, the standard deviation dropped to 23.5 daN and 21.3 daN on the left and right rollers, respectively, compared to 120.8 daN and 129.8 daN before modernization. The corresponding coefficients of variation (CVs) fell from 39.17% and 40.67% to 8.42% and 7.64%, representing an approximate 80% reduction in relative variability. A new Bootstrap analysis with 10,000 samples was performed, yielding the distribution shown in Figure 10 and statistical parameters shown in Table 4.

Comparing the two Bootstrap analyses (preliminary vs. upgraded system with old and new electronic pads, respectively), the confidence interval decreased by 3.33% for the loaded vehicle and by 73.98% for the driver-only configuration. Specifically, in the second Bootstrap, the confidence intervals were reduced to 96.67% and 26.02% of their original size, respectively. This improvement is particularly significant for the loaded vehicle test.

Post-upgrade tests were conducted across multiple days to verify short-term measurement stability, with no observable drift in braking force values under repeated conditions. However, long-term sensor stability under extended operational usage remains a subject for future investigation.

### 2.2. Validation of the fBrake Method for M_1_ and N_1_ Vehicles Compared to the Reference Method for Measuring Braking Efficiency at Full Load, Following the UNE 26110 Standard

This section details the results of the intercomparison test carried out to evaluate whether the test method employed by the fBrake system can be considered equivalent to the established reference method for measuring braking efficiency at full load. The validation was performed on vehicles of categories M_1_ and N_1_, in accordance with the guidelines of the UNE 26110 standard (March 2018; Road vehicles. Criteria for evaluating the equivalence of braking efficiency methods with respect to those defined in ISO 21069). The guidelines of UNE-EN ISO/IEC 17043 [19] and ISO 13528:2022 [20] were followed.

The vehicle families selected to validate the model are those presented in the segmentation described in the previous study [11]. Additionally, the minivan family was included; this family is very similar to that of vans, as shown in the aforementioned publication, but with a different criterion for calculating the axle mass distribution.

To determine the braking forces or braking efficiency, each wheel is measured individually. For each selected vehicle, a braking test is conducted in two scenarios, at least five times, under repeatability conditions:Scenario 1: vehicle with all seats occupied;Scenario 2: vehicle in a driver-only configuration.

The data for braking forces and braking efficiency with the vehicle loaded to its Maximum Authorized Mass (MAM) are obtained using two different approaches, one for each scenario.

Based on the braking force values determined for each test, the braking efficiency is calculated according to Directive 2014/45/UE with Equation (8) in this document, as well as its uncertainty. The mass to which the efficiency is referenced is the MAM of each vehicle.(8)$$E\left(\%\right)=\frac{F}{m\cdot g}100$$

Given the low number of tests, the measurement uncertainty is determined using the resampling technique known as Bootstrapping, analogous to the comparison described in Section 2.1.2 in the repeatability study of tests performed on the RBT. This method is prescribed by the UNE 26110 standard and allows for the generation of additional information through randomly conducted simulations. This makes it possible to determine the measurement uncertainty with an appropriate confidence level, despite the use of a small initial sample.

Using the Bootstrapping method, 10,000 values are obtained, from which the mean braking force per wheel is calculated, along with the upper and lower limits for a 95% confidence level:Lower limit: Lim_250_;Upper limit: Lim_9750_.

The combined uncertainty U_1_(E) in Equation (9) is calculated by partially deriving E with respect to each variable and combining the contributions [21]:(9)$$U_{1}\left(E\right)=\sqrt{{\left(\frac{\partial E}{\partial F}\right)}^{2}U_{F}^{2}+{\left(\frac{\partial E}{\partial MAM}\right)}^{2}U_{MAM}^{2}}\;$$
where the partial derivative of efficiency E with respect to braking force F is:(10)$$\frac{\partial E}{\partial F}=\frac{100}{MAM\cdot g}$$

And the partial derivative of efficiency E with respect to the MAM is:(11)$$\frac{\partial E}{\partial MAM}=-\frac{100\cdot F}{{\left(MAM\right)}^{2}\cdot g}$$

Substituting and simplifying, we obtain:(12)$$U_{1}\left(E\right)=\frac{100}{MAM\cdot g}\sqrt{U_{F}^{2}+{\left(\frac{F}{MAM}\right)}^{2}U_{MAM}^{2}}\;$$

Here,

U_F_ is the uncertainty associated with the measurement of the force by the brake tester used for the test (Appendix A) = 78 N.

U_MAM_ is the measurement uncertainty of the weighing scales = 20 kg following calibration standards of the scales used. For vehicles with lower MAM values, such as compact M1 vehicles, this component contributes a proportionally larger share of the total uncertainty, and its impact was accounted for in the combined uncertainty calculation.

On the other hand, U_2_ is the uncertainty of the efficiency associated with the repeatability of the force measurement obtained through Bootstrapping:(13)$$U_{2}=\frac{100}{MAM\cdot g}\frac{{Lim}_{9750}-{\;Lim}_{250}}{2}$$

The combined uncertainty U is expressed by combining uncertainties U_1_ and U_2_:(14)$$U=\sqrt{{U_{1}}^{2}+{U_{2}}^{2}}\;$$

#### Performance Evaluation

To evaluate the compatibility of methods, the figure of merit known as the normalized error (E_n_) is used, calculated as follows:(15)$$E_{n}=\frac{\left|E_{Scenario\;1}\;-{\;E}_{Scenario\;2}\right|}{\sqrt{{U_{Scenario\;1}}^{2}+{U_{Scenario\;2}}^{2}}}\;$$

### 2.3. Resulting Braking Efficiencies and Analysis

This section presents the five calculated braking efficiencies that appear in the validation results:

E_trad_: The traditional braking efficiency, calculated using the braking forces measured by the roller brake tester in the given load condition, compared to the curb weight + driver Mass in Running Order (MRO) recorded on the vehicle’s technical inspection card (ITV card).

E_insp_mass_: Braking efficiency calculated by comparing the same forces to the actual inspection mass, measured with calibrated scales at the inspection station.

E_MAM_: The efficiency that would be reported if the vehicle in inspection were directly referenced to the Maximum Authorized Mass (MAM) but actually loaded only to the inspection mass, representing an intermediate condition between the MRO and MAM.

E_equi_: The most conservative calculation, involving testing only the front axle and assuming equal μ (ideal brake-force distribution) between the front and rear axles. This approach is safety oriented but, due to the geometry of roller brake testers and multiple experiments, it has been demonstrated that the actual adhesion differs between axles, with the rear axle generally exhibiting slightly higher adhesion. This does not imply wheel lockup in a dynamic track test but results from the test geometry for light two-axle vehicles with electronic brake force distribution on an RBT.

E_fBrake_: The efficiency obtained using the custom-designed fBrake algorithm, which simulates the braking forces as if the vehicle were fully loaded, requiring both axles inspected, based on measured forces at the inspection mass. This efficiency is derived from two distinct load conditions between the MRO and MAM to ensure sufficient differentiation.

## 3. Results

The validation results for fBrake are presented for the sedan, SUV, van, and minivan segments. The text describes the results of the intercomparison test performed to assess whether the test method for the determination of the braking efficiency of a vehicle, using the fBrake, can be considered equivalent to the measurement method established as a reference for the measurement of braking efficiency at full load by the UNE 26110 standard of March 2018: “Road vehicles. Criteria for the evaluation of the equivalence of braking efficiency methods with respect to those defined in ISO 21069”.

### 3.1. Braking Efficiency Data for Model Validation

The results include the five braking efficiency calculations defined above.

#### 3.1.1. Sedan Segment: 2010 Renault Laguna

For the sedan segment validation, a 2010 Renault Laguna was used in the tests. Table 5 presents the vehicle’s key mass and dimensional parameters. The corresponding braking forces and apparent friction coefficients are provided in Table 6, while Table 7 summarizes the calculated brake efficiencies.

The fBrake efficiencies achieved with fBrake are shown on Table 7.

#### 3.1.2. SUV Segment: 2015 Nissan Juke

For the SUV segment validation, a 2015 Nissan Juke was used in the tests. Table 8 presents the vehicle’s key mass and dimensional parameters. The corresponding braking forces and apparent friction coefficients are provided in Table 9, while Table 10 summarizes the calculated brake efficiencies.

The fBrake efficiencies achieved with fBrake for the SUV validation vehicle are shown on Table 10.

#### 3.1.3. Van Segment: 2001 Mercedes-Benz Sprinter

For the van segment validation, a 2001 Mercedes-Benz Sprinter was used in the tests. Table 11 presents the vehicle’s key mass and dimensional parameters. The corresponding braking forces and apparent friction coefficients are provided in Table 12, while Table 13 summarizes the calculated brake efficiencies.

The fBrake efficiencies achieved with fBrake for the van validation vehicle are shown on Table 13.

#### 3.1.4. Minivan Segment: 2017 Kia Carens

For the minivan segment validation, a 2017 Kia Carens was used in the tests. Table 14 presents the vehicle’s key mass and dimensional parameters. The corresponding braking forces and apparent friction coefficients are provided in Table 15, while Table 16 summarizes the calculated brake efficiencies.

The fBrake efficiencies achieved with fBrake for the minivan validation vehicle are shown on Table 16.

### 3.2. Model Validation According to UNE 26110

The validation was conducted with one representative vehicle from each segment. Braking efficiency was calculated by simulation and direct measurement. The measurement uncertainties, normalized error, and both estimates were found to be compatible with normalized error values below 1 (Table 17).

## 4. Discussion

The results presented in the previous section demonstrate the feasibility of the fBrake method to reliably estimate braking efficiency at the Maximum Authorized Mass (MAM) using measurements obtained under different load conditions. This addresses a critical practical limitation in periodic technical inspections (PTIs): the impossibility of testing vehicles fully loaded, which complicates the correct implementation of Directive 2014/45/EU. The findings align directly with the objectives of the Special Issue on Advanced Sensing and Analysis Technology in Transportation Safety by showcasing how enhanced measurement and data processing methods can improve the safety and reliability of road vehicle inspections.

This investigation confirms the effectiveness of the fBrake method for calculating braking efficiency in light vehicles of categories M_1_ (passenger cars with up to nine seats, including the driver) and N_1_ (light commercial vehicles with a maximum mass not exceeding 3500 kg). The comparative experimental design included two principal scenarios:Braking tests on the same vehicles loaded to a condition close to their MAM, representing the ideal but impractical direct measurement;Braking tests on a roller brake tester with vehicles in a typical “driver-only” configuration, which reflects standard practice during regular PTIs.

To obtain the necessary data for the fBrake method, tests were planned and executed using dozens of vehicles across five segments (sedan, SUV, van, pick-up, and minivan). These were made available by volunteer owners and cooperating PTI centers. For each load configuration, five repeated tests were performed, and a Bootstrap resampling method with 10,000 samples was applied to robustly estimate the general braking behavior and quantify measurement uncertainty. Bootstrap was selected in accordance with UNE 26110 and ISO 13528 for its suitability in small-sample, data-driven scenarios. Although Monte Carlo simulation could offer an alternative for propagating model-based uncertainty, its advantages are limited in purely empirical, sample-limited PTI test environments and may be explored in future studies. The fBrake method was validated on four different vehicle segments (sedan, SUV, van and minivan). The validation of the pick-up segment is pending for future work.

All the validations performed were satisfactory. The validation outcomes indicate that the fBrake method provides results that are statistically compatible with those obtained from direct measurements under near-MAM conditions, with normalized errors consistently below the threshold defined by UNE 26110, obtaining normalized errors (E_N_) always less than 1. This demonstrates that fBrake is a practical, cost effective, and safety-compliant solution that inspection centers can adopt to better assess braking performance without the logistical challenges of loading vehicles.

Moreover, the methodology highlights the crucial role of precise sensing, robust calibration, and advanced data analysis techniques, areas that are central to ongoing improvements in vehicle safety monitoring and regulatory compliance.

While the validation results support the reliability of the fBrake method across several vehicle segments, some limitations must be acknowledged. First, the method relies on assumptions regarding load distribution and axle balance that, while generally representative, may not capture atypical or non-uniform loading scenarios. Additionally, although fBrake estimates braking efficiency at the Maximum Authorized Mass (MAM) through force measurements under alternative load conditions, it does not replace the need for a complete inspection of the braking system’s mechanical condition. PTI personnel must continue to verify system integrity, check for leaks, and inspect for signs of degradation or uneven wear, as these factors are not captured through force data alone.

The fBrake method has been tested using different RBTs and in collaboration with multiple inspection centers, and its implementation is not dependent on a specific bench model. However, future research could extend validation to a broader variety of vehicle layouts and operational contexts, especially in countries with differing PTI protocols or regulatory interpretations. While the current uncertainty model includes scale-based mass measurement errors and accounts for repeatability through Bootstrapping, additional variability sources could also be considered.

In particular, future work will explore how the coefficient of friction (μ) between tire and roller varies with vertical load and how the mechanical design and geometry of different RBT platforms may introduce axle-dependent variations in apparent μ. A dedicated study is planned to assess the influence of these factors on measurement accuracy and their implications for extending fBrake to more complex vehicle geometries or variable inspection setups.

The fBrake method is particularly well suited for integration into current PTI practices involving M1 and N1 vehicles. Directive 2014/45/EU requires that braking efficiency be referenced to the Maximum Authorized Mass (MAM), but this is rarely achieved in practice due to the operational difficulty of testing light vehicles in fully laden conditions. As a result, many EU Member States perform brake tests with vehicles in unladen or partially loaded configurations and reference the results to the Mass in Running Order (MRO). This practice does not comply with the Directive and, if applied directly using MAM as the reference without a corresponding load condition, may produce misleading results and a high rate of unjustified rejections.

Implementing fBrake as an alternative method enables compliance with the spirit of the Directive while avoiding the impracticality of requiring vehicle owners to arrive fully loaded. We recommend that inspection stations consider adopting fBrake under the conditions defined in this study, particularly for M1 and N1 vehicles, using certified RBT equipment and visual verification of brake system condition. Full-scale adoption will become more viable when future updates to the Directive formally allow equivalent estimation methods for these vehicle categories, as is currently allowed for heavier vehicles.

## Figures and Tables

**Figure 1 sensors-25-04522-f001:**
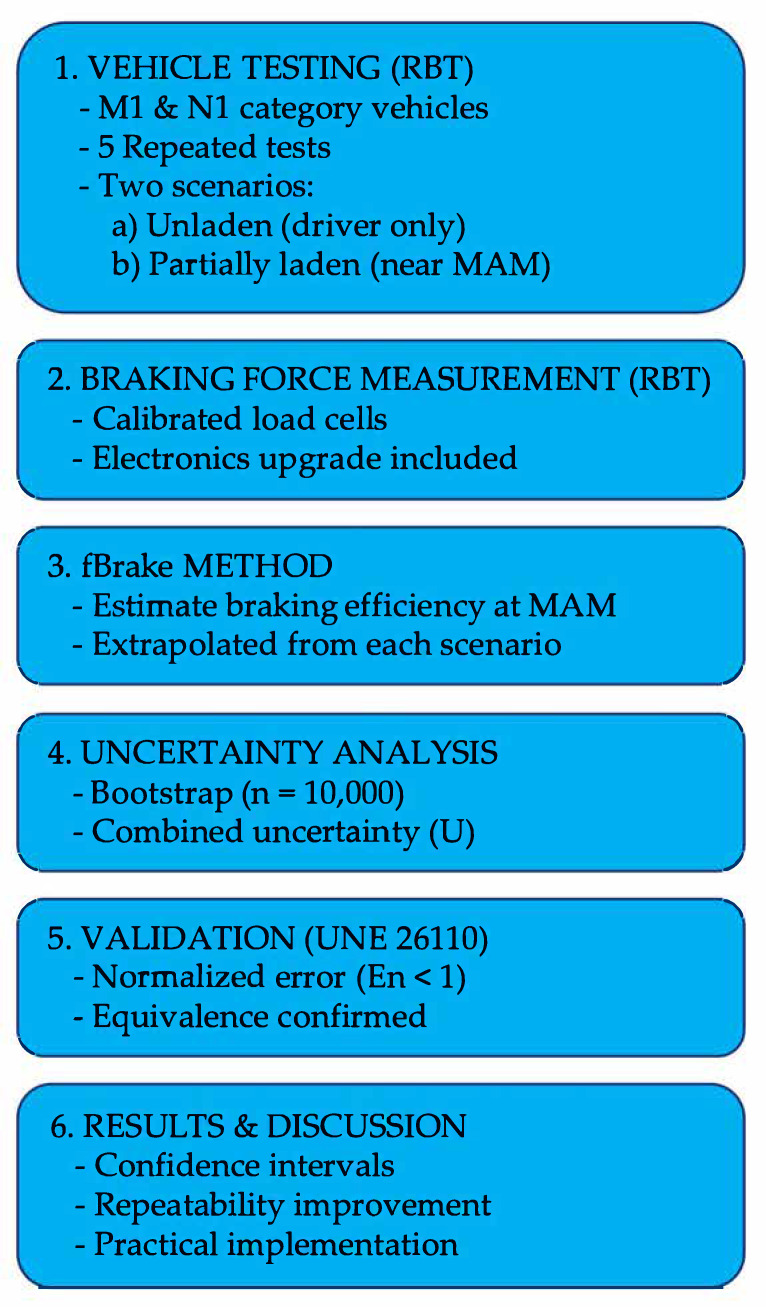
General methodology roadmap for the experimental validation of the fBrake method. The diagram summarizes the main phases of the study, including vehicle testing under two load conditions, brake force measurement via a roller brake tester (RBT), the application of the fBrake estimation model, uncertainty analysis using the Bootstrap method, and the final equivalence assessment following UNE 26110 standard criteria.

**Figure 2 sensors-25-04522-f002:**
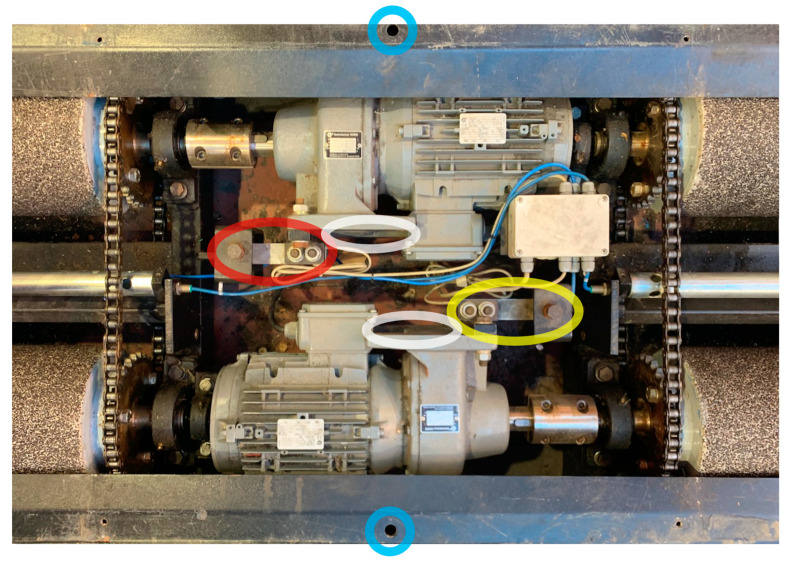
Location of load cells (red and yellow), anchorages for the calibrated load cell (white), and anchorages for the calibration tool created to calibrate the Capatest roller brake tester, model RBT-2000/F (blue).

**Figure 3 sensors-25-04522-f003:**
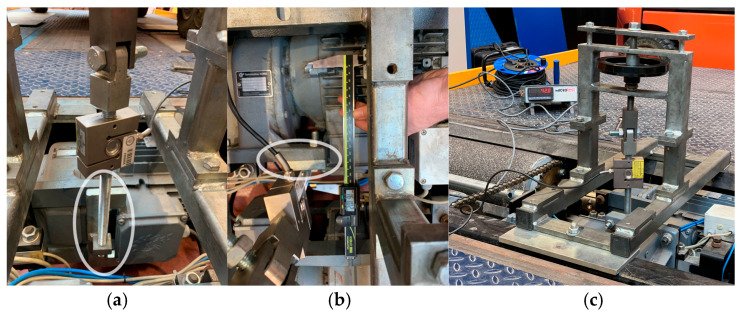
Calibration of the load cells of the Capatest model RBT-2000/F brake tester. (**a**,**b**) Anchorages for the anchorages for the calibrated load cell (white ellipse). (**c**) Calibration tool for both tension and compression load cells.

**Figure 4 sensors-25-04522-f004:**
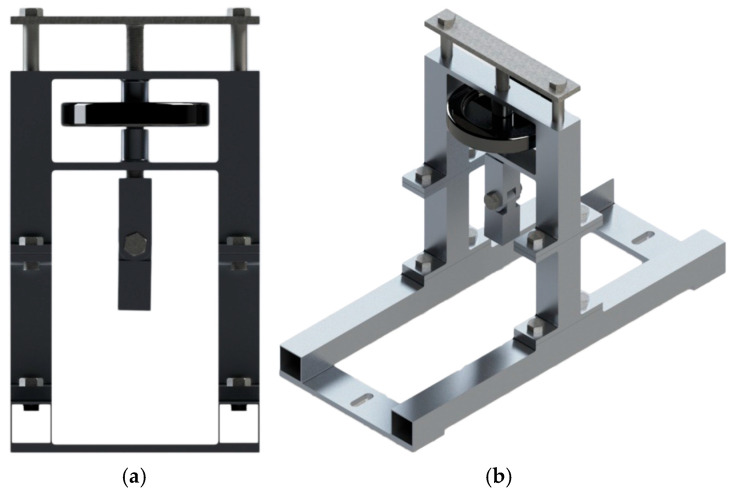
The 3D-modeleded tool to calibrate the RBT. (**a**) Front view. (**b**) Isometric pictorial.

**Figure 5 sensors-25-04522-f005:**
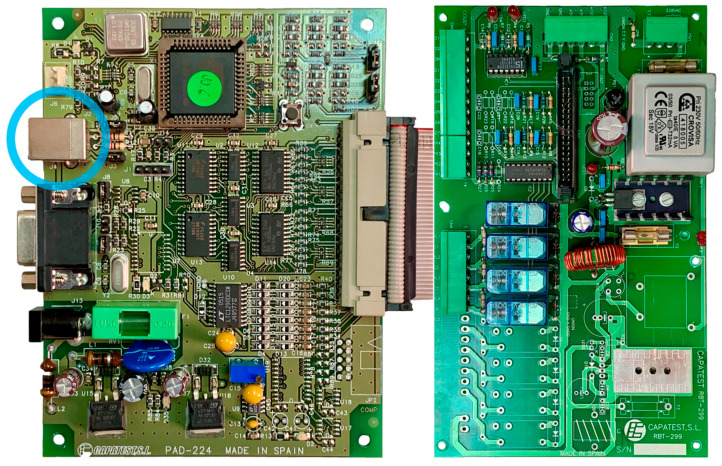
Electronic boards RBT 299 + PAD 224. USB type B port highlighted in light blue.

**Figure 6 sensors-25-04522-f006:**
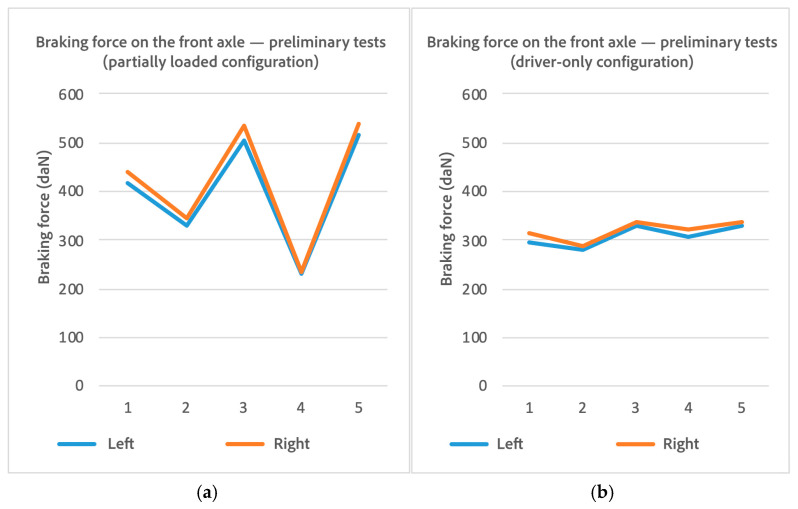
Braking force in 5 different preliminary tests: (**a**) partially loaded vehicle; (**b**) vehicle in driver-only configuration.

**Figure 7 sensors-25-04522-f007:**
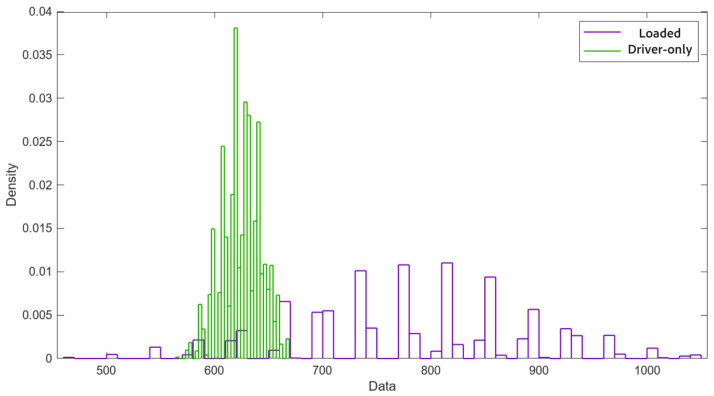
Bootstrapping normal distribution. Loaded vehicle vs. driver-only configuration. Old electronics (data: daN, front axle).

**Figure 8 sensors-25-04522-f008:**
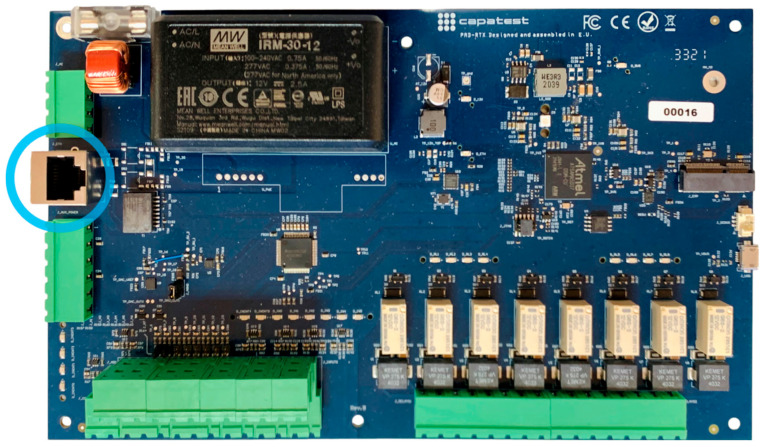
New electronic board PAD-728. Ethernet port highlighted in blue.

**Figure 9 sensors-25-04522-f009:**
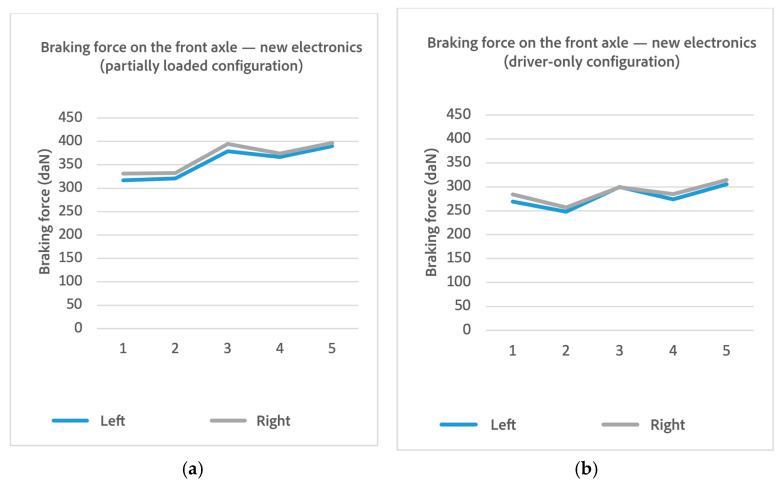
Braking force in 5 different tests with the new electronics: (**a**) partially loaded vehicle; (**b**) vehicle in driver-only configuration.

**Figure 10 sensors-25-04522-f010:**
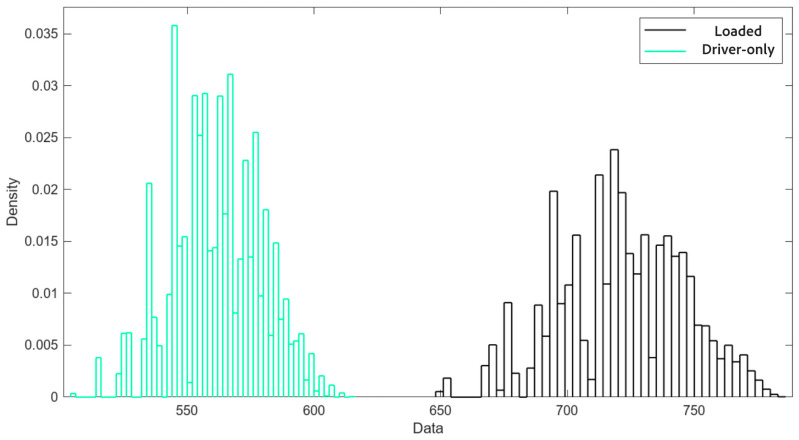
Bootstrapping normal distribution. Loaded vehicle vs. driver-only configuration. New electronics (data: daN, front axle).

**Table 1 sensors-25-04522-t001:** Preliminary test of VW Scirocco on RBT-2000/F brake tester (front axle). Electronics PAD-224/RBT-299 and PC with Windows XP.

	Unloaded			Loaded		
	Left (daN)	Right (daN)	Difference (%)	Left (daN)	Right (daN)	Difference (%)
Test 1	416	441	6.01	296	314	6.08
Test 2	328	343	4.57	279	287	2.87
Test 3	503	533	5.96	331	337	1.81
Test 4	230	234	1.74	308	320	3.90
Test 5	516	537	4.07	328	338	3.05
Average	398.6	417.6	4.47	308.4	319.2	3.54
SD	120.8	129.8	1.75	21.87	20.83	1.60

**Table 2 sensors-25-04522-t002:** Preliminary front axle test data after Bootstrap of 10,000 samples using RBT 299 + PAD 224 electronics and PC with Windows XP.

	Driver-Only Config.	Partially Loaded
Average (daN)	619.6	701.8
SD (daN)	17.86	104.1
Upper limit (daN) (95% confidence)	659.2	969.8
Lower limit (daN) (95% confidence)	587.2	584.0

**Table 3 sensors-25-04522-t003:** Test after RBT electronic modernization with VW Scirocco (front axle), with PAD-728 and new PC with Windows 10.

	Unloaded			Loaded		
	Left (daN)	Right (daN)	Difference (%)	Left (daN)	Right (daN)	Difference (%)
Test 1	317	331	4.42%	269	284	5.58%
Test 2	321	332	3.55%	248	257	3.55%
Test 3	379	395	4.17%	300	299	−0.33%
Test 4	367	374	2.02%	274	284	3.80%
Test 5	390	397	1.85%	305	314	3.08%
Average	354.8	366.0	3.15%	279.2	279	−0.11%
SD	33.7	32.5	1.20%	23.5	21.3	2.16%

**Table 4 sensors-25-04522-t004:** Front axle test data after Bootstrap of 10,000 samples using RBT RTX electronics and new Windows 10 PC.

	Driver-Only Config.	Partially Loaded
Average (daN)	561.8	721.2
SD (daN)	17.71	25.07
Upper limit (daN) (95% confidence)	594.8	770.0
Lower limit (daN) (95% confidence)	525.2	669.6

**Table 5 sensors-25-04522-t005:** Mass and dimensional data. Sedan segment for validation.

	Front Axle	Rear Axle	Total
Mass in Scenario 2 (kg)	1030	593	1623
Mass in Scenario 1 (kg)	1120	788	1908
GVWR (MAM) (kg)	1220	1030	2025
Curb weight + driver (MRO) (kg)	-	-	1555
Wheelbase (mm)	-	-	2756
Total height (mm)	-	-	1445

**Table 6 sensors-25-04522-t006:** Braking forces after Bootstrapping and apparent friction coefficient (μ): Sedan.

	Front Axle	Rear Axle
F_x,Scenario2_ (N)	6598	4801
F_x,Scenario1_ (N)	7349	6530
Apparent μ, Scenario 2	0.654	0.826
Apparent μ, Scenario 1	0.670	0.846

**Table 7 sensors-25-04522-t007:** fBrake efficiencies achieved in both scenarios. Sedan segment for validation.

Method	Efficiency (Scenario 2)	Efficiency (Scenario 1)
E_trad_	74.80%	91.08%
E_insp_mass_	71.67%	74.23%
E_MAM_	57.44%	69.94%
E_equi_	62.78%	64.30%
E_fBrake_	74.82%	76.23%

**Table 8 sensors-25-04522-t008:** Mass and dimensional data. SUV segment for validation.

	Front Axle	Rear Axle	Total
Mass in Scenario 2 (kg)	860	490	1350
Mass in Scenario 1 (kg)	915	670	1585
GVWR (MAM) (kg)	985	830	1770
Curb weight + driver (MRO) (kg)	-	-	1370
Wheelbase (mm)	-	-	2530
Total height (mm)	-	-	1565

**Table 9 sensors-25-04522-t009:** Braking forces after Bootstrapping and apparent friction coefficient (μ): SUV.

	Front Axle	Rear Axle
F_x,Scenario2_ (N)	5036	3367
F_x,Scenario1_ (N)	5256	4869
Apparent μ, Scenario 2	0.597	0.701
Apparent μ, Scenario 1	0.573	0.742

**Table 10 sensors-25-04522-t010:** fBrake efficiencies achieved in both scenarios. SUV segment for validation.

Method	Efficiency (Scenario 2)	Efficiency (Scenario 1)
E_trad_	62.59%	75.41%
E_insp_mass_	63.51%	64.78%
E_MAM_	48.44%	58.37%
E_equi_	59.75%	55.69%
E_fBrake_	64.99%	65.72%

**Table 11 sensors-25-04522-t011:** Mass and dimensional data. Van segment for validation.

	Front Axle	Rear Axle	Total
Mass in Scenario 2 (kg)	1370	1116	2486
Mass in Scenario 1 (kg)	1720	1742	3462
GVWR (MAM) (kg)	1708	1792	3500
Curb weight + driver (MRO) (kg)	-	-	2183
Wheelbase (mm)	-	-	3556
Total height (mm)	-	-	2625

**Table 12 sensors-25-04522-t012:** Braking forces after Bootstrapping and apparent friction coefficient (μ): Van.

	Front Axle	Rear Axle
F_x,Scenario2_ (N)	7878	5394
F_x,Scenario1_ (N)	9545	8272
Apparent μ, Scenario 2	0.587	0.493
Apparent μ, Scenario 1	0.566	0.485

**Table 13 sensors-25-04522-t013:** fBrake efficiencies achieved in both scenarios. Van segment for validation.

Method	Efficiency (Scenario 2)	Efficiency (Scenario 1)
E_trad_	62.04%	83.28%
E_insp_mass_	58.48%	52.51%
E_MAM_	38.69%	51.94%
E_equi_	56.35%	54.38%
E_fBrake_	52.15%	52.54%

**Table 14 sensors-25-04522-t014:** Mass and dimensional data. Minivan segment for validation.

	Front Axle	Rear Axle	Total
Mass in Scenario 2 (kg)	1370	1116	2486
Mass in Scenario 1 (kg)	1720	1742	3462
GVWR (MAM) (kg)	1708	1792	3500
Curb weight + driver (MRO) (kg)	-	-	2183
Wheelbase (mm)	-	-	3556
Total height (mm)	-	-	2625

**Table 15 sensors-25-04522-t015:** Braking forces after Bootstrapping and apparent friction coefficient (μ): Minivan.

	Front Axle	Rear Axle
F_x,Scenario2_ (N)	7878	5394
F_x,Scenario1_ (N)	9545	8272
Apparent μ, Scenario 2	0.587	0.493
Apparent μ, Scenario 1	0.566	0.485

**Table 16 sensors-25-04522-t016:** fBrake efficiencies achieved in both scenarios. Minivan segment for validation.

Method	Efficiency (Scenario 2)	Efficiency (Scenario 1)
E_trad_	74.54%	90.29%
E_insp_mass_	68.35%	70.47%
E_MAM_	54.34%	65.82%
E_equi_	52.90%	52.98%
E_fBrake_	73.49%	74.60%

**Table 17 sensors-25-04522-t017:** fBrake validation for the four selected segments.

Vehicle	E_Scenario 2_ (%)	U_Scenario 2_ (%)	E_Scenario 1_ (%)	U_Scenario 1_ (%)	E_N_ (%)
Sedan	74.82	0.94	76.23	0.90	0.77
SUV	64.99	0.92	65.72	1.18	0.35
Van	52.15	0.65	52.54	0.67	0.30
Minivan	73.49	1.35	74.60	1.09	0.46

## Data Availability

The original contributions presented in the study are included in the article; further inquiries can be directed to the corresponding author.

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
