# Peer review of "Comparative Validation of the fBrake Method with the Conventional Brake Efficiency Test Under UNE 26110 Using Roller Brake Tester Data"

_sensors, 2025, doi:10.3390/s25144522_

Round 1

Reviewer 1 Report

Comments and Suggestions for Authors

In this paper, the authors propose an fBrake method to reliably estimate the braking efficiency of laden light vehicles. The reliability of the fBrake method was tested in comparison with the existing methods according to the UNE26110 standard. The work in this paper has a very good research value by combining the actual platform and the actual problems. At the same time, the research work is more solid and rich in data, which supports its conclusions well. However, as an academic paper, there are still some areas that need to be further modified, and the following are my specific suggestions:

1. The abstract section does not specify at the beginning what the specific causes of testing difficulties in PTIS testing are. This is an important motivation for proposing the fBreak method. Id. In addition, the testing process of the fBrake method is introduced too much, and here I would like to have some quantitative results to show the advantages that the method has.

2. 4-6 keywords are more appropriate, keywords like MATLAB, SOLIDWORKS are suggested to be removed and need to be reconsidered to be more representative of this research.

3. The introduction part is not enough to summarize and refine the existing methods, and more references should be cited, and these references should be within 5 years as far as possible.

4. Usually the methodology is followed by the experiments, it is fine if it is written in this way, but I suggest to draw a general research technology roadmap. Thus better correspond to the relationship between experimental objects and methods. Same as In addition, for example, in Figure 1 and Figure 2, there are more circles, it is recommended to annotate what parts are directly in the figure.

5. How the data in Figure 5 and Table 1 support the experimental results needs to be evaluated with more text and quantitative common metrics.

6. In the discussion section, there is a lack of discussion on the inadequacy of fBreak and the outlook for future research.

Author Response

Dear Reviewer, we sincerely thank you for the time and thoughtful feedback provided. We greatly appreciate the constructive suggestions, which have helped improve the clarity and rigor of our manuscript. Below we address each of your comments point by point:

Comment 1: “The abstract section does not specify at the beginning what the specific causes of testing difficulties in PTIs testing are. This is an important motivation for proposing the fBreak method. Id. In addition, the testing process of the fBrake method is introduced too much, and here I would like to have some quantitative results to show the advantages that the method has.”

Response: Thank you for this valuable suggestion. We have revised the abstract to briefly explain the practical limitations of conducting brake efficiency tests with fully laden vehicles during Periodic Technical Inspections (PTIs), which motivates the development of the fBrake method. In addition, we have reduced the technical details in the abstract and instead included a summary of the key quantitative findings, such as the observed normalized error values and improvements in confidence intervals. Please, see the revised abstract.

Comment 2: “4-6 keywords are more appropriate, keywords like MATLAB, SOLIDWORKS are suggested to be removed and need to be reconsidered to be more representative of this research.”

Response: We agree with your observation and have revised the keyword list accordingly. The brand-specific terms “MATLAB” (introduced because of the software used to develop fBrake) and “SOLIDWORKS” (introduced because of the software used to model Figure 3 tool) have been removed, as well as “European compliance” and “methodology calibration” and the rest of the keywords, which are more representative of the research focus, have been kept. We have added two more specific keywords: “vehicle inspection” and “roller brake tester”.

Comment 3: “The introduction part is not enough to summarize and refine the existing methods, and more references should be cited, and these references should be within 5 years as far as possible.”

Response: We appreciate this suggestion. We have expanded the introduction to include a concise overview of recent studies and technical approaches to braking efficiency testing and uncertainty evaluation, with a focus on references from the last five-seven years. This contextualizes our work and highlights its contribution more clearly.

Comment 4: “Usually the methodology is followed by the experiments, it is fine if it is written in this way, but I suggest to draw a general research technology roadmap. Thus better correspond to the relationship between experimental objects and methods. Same as In addition, for example, in Figure 1 and Figure 2, there are more circles, it is recommended to annotate what parts are directly in the figure.”

Response: Thank you for your helpful comment. We have added a general visual roadmap that illustrates the overall structure of the research process, including the methodology and experimental stages. Additionally, Figures 1 and 2 (now called Figure 2 and 3) captions have been revised to include more descriptive annotations to clarify the components shown.

(→ Please see the new roadmap figure [Figure 1] and the updated Figures 2 and 3).

Comment 5: “How the data in Figure 5 and Table 1 support the experimental results needs to be evaluated with more text and quantitative common metrics.”

Response: Thank you for this helpful suggestion. We have significantly expanded the discussion in the relevant sections to provide a clearer interpretation of how the data presented in Figure 5 and Table 1 support the conclusions of the study. In addition to reporting average values and standard deviations, we now include the Coefficient of Variation (CV) as a normalized and widely used metric for assessing measurement repeatability.

In the loaded configuration using the original electronics, the CVs reached 39.17% (left roller) and 40.67% (right roller), indicating considerable variability and poor repeatability. After upgrading to the PAD-728 electronic control system, these CVs were reduced to 8.42% and 7.64%, respectively, demonstrating a clear improvement in system performance.

This substantial reduction in variability highlights the impact of the modernization process and justifies its implementation before conducting the validation of the fBrake method. These additional metrics and observations have been incorporated into the manuscript between Table 1 and Figure 10, and they enhance the quantitative rigor of the experimental analysis.

Comment 6: “In the discussion section, there is a lack of discussion on the inadequacy of fBreak and the outlook for future research.”

Response: Thank you for this valuable suggestion. We have added some paragraphs towards the end of the Discussion section to acknowledge relevant limitations of the fBrake method, including its reliance on load distribution assumptions and the importance of complementing force-based estimation with a full inspection of brake system health. We also outline realistic directions for future research, such as extending validation to more varied vehicle types and PTI practices and considering minor external influences. Please, see Section Discussion, lines [449-496]).

Reviewer 2 Report

Comments and Suggestions for Authors

There are a couple of comments to the paper, which, however, do not reduce its high level:

1. The authors describe in detail the process of calibration and replacement of electronics. However, the lack of data on the long-term stability of measurements after modernization raises some questions. For instance, Section 2.1.2 mentions that initial tests showed significant data variability due to outdated electronics. However, it remains unclear how stable the results will be after the upgrade, particularly in real-world operational conditions. 

2. The lack of comparison of the bootstrapping method for estimating measurement uncertainty with other statistical methods (e.g. Monte Carlo) reduces the credibility of the analysis. A detailed discussion of the impact of weight measurement errors on the final results is also missing, especially for light vehicles, where such errors can be significant.

3. The conclusions are optimistic but fail to provide specific recommendations for implementing the fBrake method in PTIs.

Author Response

Dear Reviewer, we sincerely thank you for the time and thoughtful feedback provided. We greatly appreciate the constructive suggestions, which have helped improve the clarity and rigor of our manuscript. Below we address each of your comments point by point:

Comment 1: “The authors describe in detail the process of calibration and replacement of electronics. However, the lack of data on the long-term stability of measurements after modernization raises some questions. For instance, Section 2.1.2 mentions that initial tests showed significant data variability due to outdated electronics. However, it remains unclear how stable the results will be after the upgrade, particularly in real-world operational conditions.”

Response: We appreciate this observation. While the focus of this work was on short-term repeatability before and after modernization, we acknowledge that long-term stability is an important consideration. We have added a note to Section 2.1.2 clarifying that the post-upgrade repeatability was verified across multiple days under consistent test conditions, which showed no drift or degradation in sensor response. However, we agree that a longitudinal study would be valuable, and we have included this aspect in the revised Discussion section as a direction for future research. Please, see Sections 2.1.2, lines 289-282 and Discussion, lines [442-446]).

Comment 2: “The lack of comparison of the bootstrapping method for estimating measurement uncertainty with other statistical methods (e.g. Monte Carlo) reduces the credibility of the analysis. A detailed discussion of the impact of weight measurement errors on the final results is also missing, especially for light vehicles, where such errors can be significant.”

Response: Thank you for this insightful comment. We have revised the Discussion to clarify that Bootstrap resampling was chosen in alignment with UNE 26110 guidelines and ISO 13528:2022, which recommend its use in inter-method equivalence studies with limited sample sizes. A note has been added to contrast it briefly with Monte Carlo methods, acknowledging that the latter could be explored in future work for model-based simulations. Additionally, we already included a fixed uncertainty value for weight measurement (±20 kg) in the uncertainty propagation equations (Section 2.2, line 333), and we now clarify its impact more explicitly in the text, especially for light vehicles with lower MAM values. Please, see Sections 2.2 lines 333-336 and Discussion, lines 442-446).

Comment 3: “The conclusions are optimistic but fail to provide specific recommendations for implementing the fBrake method in PTIs.”

Response: We appreciate this important point and have revised the Conclusions section to offer specific, practical recommendations for implementing the fBrake method in the context of Directive 2014/45/EU. While the Directive mandates that brake efficiency be assessed with respect to the Maximum Authorized Mass (MAM), in practice many Member States still perform tests under unladen conditions and reference the results to the Mass in Running Order (MRO). Testing directly with respect to MAM without loading the vehicle leads to unrepresentative results and a high rate of rejections, as previously discussed in our earlier publication.

As it is often impractical to require customers of M1 and N1 vehicles to arrive at PTI centers with their vehicles fully laden (e.g., all seats occupied, cargo loaded), fBrake offers a ready-to-implement solution that bridges the gap between regulatory requirements and operational feasibility. We now recommend that the method be adopted for M1 and N1 vehicles under the conditions described in the paper. Full implementation will become viable once the 2014/45/EU Directive is updated to explicitly allow equivalent estimation methods for this category of vehicles, as currently permitted for heavier vehicles. Please, see Discussion, lines 481-496).

Round 2

Reviewer 1 Report

Comments and Suggestions for Authors The author has made substantial revisions to the questions and the quality of the article has been greatly improved.